✧ PLOS | ONE

# Insights into the hyperglycosylation of human chorionic gonadotropin revealed by glycomics analysis

Linda Ibeto[1,2], Aristotelis Antonopoulos[1], Paola Grassi[1], Poh-Choo Pang[1],
Maria Panico[1], Shabnam Bobdiwala[3], Maya Al-Memar[3], Paul Davis[4], Mark Davis[4],
Julian Norman Taylor[2], Paula Almeida[2], Mark R. Johnson[2], Richard Harvey[5],
Tom Bourne[3], Michael Seckl[5], Gary Clark[6], Stuart M. Haslam[1], Anne Dell[1] *

1 Department of Life Sciences, Imperial College London, London, United Kingdom, 2 Academic Department
of Obstetrics and Gynaecology, Chelsea and Westminster Hospital, London, United Kingdom, 3 Tommys'
National Centre for Miscarriage Research, Queen Charlottes' & Chelsea Hospital, Imperial College, London,
United Kingdom, 4 Mologic LTD, Bedford Technology Park, Bedfordshire, United Kingdom, 5 Division of
Cancer, Department of Surgery and Cancer, Imperial College London, London, United Kingdom,
6 Department of Obstetrics, Gynaecology and Women's Health, University of Missouri, Columbia, Missouri,
United States of America

* a.dell@imperial.ac.uk

Paulo, BRAZIL

**Data Availability Statement:** All relevant data are
within the paper and its Supporting Information
files.

## Abstract

Human chorionic gonadotropin (hCG) is a glycoprotein hormone that is essential for the
maintenance of pregnancy. Glycosylation of hCG is known to be essential for its biological
activity. "Hyperglycosylated" variants secreted during early pregnancy have been proposed
to be involved in initial implantation of the embryo and as a potential diagnostic marker for
gestational diseases. However, what constitutes "hyperglycosylation" is not yet under-
stood. In this study, we perform comparative N-glycomic analysis of hCG expressed in the
same individuals during early and late pregnancy to help provide new insights into hCG
function, reveal new targets for diagnostics and clarify the identity of hyperglycosylated
hCG. hCG was isolated in urine collected from women at 7 weeks and 20 weeks' gestation.
hCG was also isolated in urine from women diagnosed with gestational trophoblastic dis-
ease (GTD). We used glycomics methodologies including matrix assisted laser desorption/
ionisation–time of flight (MALDI-TOF) mass spectrometry (MS) and MS/MS methods to
characterise the N-glycans associated with hCG purified from the individual samples. The
structures identified on the early pregnancy (EP-hCG) and late pregnancy (LP-hCG) sam-
ples corresponded to mono-, bi-, tri-, and tetra-antennary N-glycans. A novel finding was the
presence of substantial amounts of bisected type N-glycans in pregnancy hCG samples,
which were present at much lower levels in GTD samples. A second novel observation was
the presence of abundant Lewis[X] antigens on the bisected N-glycans. GTD-hCG had fewer
glycoforms which constituted a subset of those found in normal pregnancy. When compared
to EP-hCG, GTD-hCG samples had decreased signals for tri- and tetra-antennary N-gly-
cans. In terms of terminal epitopes, GTD-hCG had increased signals for sialylated struc-
tures, while Lewis[X] antigens were of very minor abundance. hCG carries the same N-
glycans throughout pregnancy but in different proportions. The N-glycan repertoire is more

**Funding:** Affiliation with Mologic LTD UK The funder provided support in the form of salaries for authors [PG], but did not have any additional role in the study design, data collection and analysis, decision to publish, or preparation of the manuscript. The specific roles of these authors are articulated in the 'author contributions' section

**Competing interests:** Affiliation with Mologic LTD UK I can confirm that this commercial affiliation does not alter our adherence to PLOS ONE policies on sharing data and materials." (as detailed online in guide for authors http://journals.plos.org/plosone/s/competing-interests).

diverse than previously reported. Bisected and Lewis[X] structures are potential targets for diagnostics. hCG isolated from pregnancy urine inhibits NK cell cytotoxicity *in vitro* at nano-molar levels and bisected type glycans have previously been implicated in the suppression of NK cell cytotoxicity, suggesting that hCG-related bisected type N-glycans may directly suppress NK cell cytotoxicity.

## Introduction

Human chorionic gonadotropin (hCG) is an essential pregnancy-associated glycoprotein that is a member of the same family that includes luteinizing hormone, follicle stimulating hormone and thyroid stimulating hormone. Each member is a heterodimer consisting of shared α-subunits in non-covalent association with distinct β-subunits that confer their specific physiological activities [1, 2]. Pregnancy associated hCG is mainly secreted by the syncytiotrophoblast associated with the embryo and is excreted in urine.

hCG levels in maternal blood increase progressively during early pregnancy, reaching maximal levels between the 8-11[th] week of gestation. Thereafter, the level of hCG declines until around twenty weeks and then remains comparatively low for the remainder of the pregnancy. The functional roles of hCG throughout pregnancy have been extensively studied [1, 3–6] and are thought to be necessary for the maintenance of pregnancy by mediating multiple placental, uterine and fetal functions [7–10]. hCG may play a part in implantation through its receptors on the endometrium [10] and is thought to aid smooth muscle relaxation, maintain quiescence of the myometrium in pregnancy and promote myometrial vasodilatation [7, 9, 11]. hCG induces relaxin secretion by the corpus luteum during the luteal phase and in early pregnancy. Both relaxin and progesterone play an important role in the maintenance of early pregnancy [9, 10]. hCG may also contribute to maternal immune tolerance [8] but as pregnancy progresses into the second and third trimester, the role of this hormone becomes less apparent. hCG is also expressed in several trophoblastic diseases and is often used as a sensitive biomarker for malignancy [12]. This hormone is expressed in both gonadal and non-gonadal tumours [13, 14].

hCG is more than 30% carbohydrate by mass, consistent with a high level of glycosylation [15]. The α-subunit bears N-linked glycans at Asn-52 and Asn-78 [16, 17]. The β-subunit is also N-glycosylated at Asn-13 and Asn-30 and in addition it bears O-glycans linked to Ser 121, 127, 132 and 138 [17, 18]. Glycosylation has been demonstrated to affect the biological activity of hCG as well as its half-life in the circulation [19]. Glycoproteins are known to be involved in several essential recognition and cell signalling events. Alterations in sugar structures have been identified as an underlying cause of malignancy [20]. The expression of specific glycoforms of hCG during pregnancy disorders and cancer could offer targets for clinical diagnosis. The N-glycans of hCG have been studied by a variety of methods including chromatography, NMR and mass spectrometry (MS) [21–24]. They have been reported to consist mainly of mono- and bi-antennary complex type structures with terminal sialic acid and with or without core fucosylation [1, 17, 24, 25]. Some glycans have been specifically associated with malignancy [26–28]. For example, increases in tri-antennary N-glycans have been linked to hCG from tumour cells [29]. Abnormal bi-antennary N-glycans have also been confirmed on tumour-derived hCG [1, 17].

Three aspects of hCG glycosylation are not well understood. First, the glycosylation of hCG from healthy pregnant women is not well defined because studies employing sophisticated

analytical methodologies have usually focused on the more abundant hCG expressed by trophoblastic tumours [27, 30]. Second, not much is known about the glycosylated forms (glycoforms) of hCG that are synthesised during obstetrical syndromes. Third, a "hyperglycosylated" variant of hCG (hCG-h) has been proposed, the nature of which is ill defined [31].

hCG-h is reported to be produced by cytotrophoblast cells while hCG is made by syncytiotrophoblast cells [3, 31]. hCG-h is specifically recognised by a monoclonal antibody called B152 which was raised using a choriocarcinoma-derived form of hCG which contained unusually high levels of branched O-glycans. [26, 29, 32, 33]. The expression 'hyperglycosylated hCG' was initially used to describe hCG containing tri-antennary N-glycans and tetra-saccharide core O-glycans when compared to the predominant bi-antennary N-glycans and di-saccharide core O-glycans (Elliott, 1997). However, following the development and use of the B152 antibody (Birkin et al., 1999), the term has come to mean hCG forms detected by assays employing this antibody (Cole, 1999). B152 binding studies have suggested that very early pregnancy forms of hCG are mainly hCG-h. The hyperglycosylation defined by B152 recognition was originally attributed to the branched O-glycans. However, structural studies have indicated that branched N-glycans contribute to the higher molecular weight of hCG-h on SDS-PAGE gels [27]. Cole et al have associated the expression of hCG-h with Down's syndrome [34]. Other early pregnancy complications including early pregnancy loss have also been linked to this isoform [32]. Controversially, it has been reported that determining the levels of hCG-h in serum and urine allows discrimination of benign from malignant gestational trophoblastic neoplasia [35]. This claim has not been subjected to rigorous physicochemical scrutiny. Likewise, the hypothesis that early pregnancy hCG (hCGh) is differentially glycosylated in comparison with hCG from later stages of pregnancy has not yet been experimentally validated.

A better understanding of hCG glycosylation could open the door to its utilisation as a biomarker for different pregnancy related conditions. Previous structural studies have focused on pooled urine samples. In the current study, samples of hCG derived from individual women were analysed by matrix assisted laser desorption ionisation–time of flight (MALDI-TOF) MS to determine if any differences between individuals exist. Attention was paid to the definition of the changes that occurred between early and late pregnancy as well as comparisons with GTD. Our aim was to rigorously analyse the N-glycans associated with hCG from normal and abnormal pregnancy in order to identify potential biomarkers and provide new insights into hCG function. Another goal was to define what actually constituted the hyperglycosylated subtype.

## Materials and methods

Consent and Ethical approval for the research project was obtained from NHS National Research Ethics Service (NRES) Riverside Committee London (REC 14/LO/0199) and the Imperial College Healthcare Tissue Bank (ICHTB, reference no. Obs-TB-13-033). All subjects gave informed consent, and patient anonymity was preserved. The urine samples used were fully anonymized and samples were accessed over between November 2013 and June 2015.

### Purification of hCG from human urine samples

Urine samples were collected from 4 healthy pregnant women during both early (weeks 6–9) and late pregnancy (weeks 20–21), i.e. matched paired samples, and from 4 patients visiting the surgical unit for evacuation of a molar pregnancy from November 2013 to June 2015 at the Early Pregnancy and Acute Gynaecology Unit (EPAGU) at Queen Charlottes' and Chelsea Hospital, a UK-based university hospital. In all 4 patients the evacuated mole was histologically confirmed as being a complete hydatidiform mole.

Urine samples were analysed for total hCG (hCG + hCGβ + other identified variant forms) using an in-house radioimmunoassay assay, which uses a polyclonal rabbit antiserum (Harvey et al. 2010). Urine samples were initially concentrated using Millipore Prep scale TFT Regenerated Cellulose 5 K Filter Cartridge (CDUF 001LC). The concentrated samples were then diluted in 0.6 M ammonium sulphate (pH 7.5) and loaded onto a Phenyl Sepharose chromatography column. Samples were firstly eluted with 50 mM ammonium bicarbonate at pH 7.5 (elutes nicked hCG) followed by a second elution with 40% (v/v) aqueous ethanol. The peak fractions from the 40% ethanol elution were pooled and exchanged into PBS, 0.1% sodium azide using GE PD10 desalting columns. Purified hCG samples were stored at -80 ºC. Proteomics was performed to assess hCG purity, representative data is presented in tables in S1 and S2 Tables.

## Analysis of purified hCG protein

Purified hCG protein samples were analysed according to established procedures [36]. Briefly, each hCG sample was reduced with 2 mg/ml DTT (in 0.6 M degassed TRIS buffer, pH 8.5, 1 h at 37 ºC) and carboxymethylated with 12 mg/ml IAA (in 0.6 M degassed TRIS buffer, pH 8.5, in the dark). Lyophilised samples were subjected to digestion with trypsin (300 µl of 50 mM ammonium bicarbonate buffer, pH 8.4) for 14 h at 37 ºC. The reaction was then terminated by incubating the digests at 100 ºC for 3 minutes. Digested hCG protein was then purified by $C_{18}$-Sep-Pak HLB plus (Waters), reduced the volume with a SpeedVac concentrator (Thermo) and lyophilised. N-glycans were released from the above lyophilised glycopeptide fractions by peptide N-glycosidase F (PNGase-F, E.C. 3.5.1.52; Roche Applied Science) digestion. Lyophilised sample was dissolved in 200 µl of ammonium bicarbonate buffer (50 mM, pH 8.4) and 10 U of PNGase-F enzyme was added to each sample and incubated at 37ºC for 24 hours. The free N-glycans were purified by Classic $C_{18}$-Sep-Pak (part number WAT051910, Waters) chromatography, reduced the volume with a SpeedVac concentrator (SPD131DDA, Thermo) and lyophilised.

Purified N-glycans were permethylated with the sodium hydroxide procedure[37]. Permethylated samples were then purified by $C_{18}$-Sep-Pak by stepwise elution with 15%, 35%, 50% and 75% of aqueous acetonitrile. Permethylated N-glycans were found in the 50% acetonitrile fraction.

## Bisected N-glycan analysis

To define N-glycan structures with bisected GlcNAc (GlcNAcβ1,4Man), galactosylation using β1,4 galactosyltransferase was carried out [38]. Lyophilised purified native N-glycans (non-permethylated N-glycans) were dissolved in 150 µl of 50 mM MOPS buffer containing 45 µM UDP-galactose (pH 7.4, adjusted with ammonia) before the addition of 20 mM manganese (II) chloride 4-hydrate. For the galactosylation enzymatic reaction, 10 µl of bovine milk β1,4 galactosyltransferase (Calbiochem) was added and it was incubated for 20 h at 37˚C and lyophilised. During the β1,4 galactosyltransferase experiment, any terminal GlcNAc residue on the α1,3 and α1,6 mannose arms of a N-glycan is modified by the addition on a galactose residue, whereas the GlcNAc terminated residue attached to the mannose of the core chitobiose structure (bisected N-glycan) is not modified by the addition of a galactose residue.

## MALDI mass spectrometry and data interpretation

The analysis of the permethylated N-glycan structures was performed employing matrix-assisted laser desorption ionisation-time of flight (MALDI-TOF MS and MALDI-TOF/TOF MS/MS) mass spectrometry. MS and MS/MS data were acquired using a 4800 MALDI-TOF/TOF

(Applied Biosystems) mass spectrometer. Permethylated samples were dissolved in 10 μl of methanol, and 1 μl of dissolved sample was premixed with 1 μl of matrix (10 mg/ml 3,4-diami-nobenzophenone in 75% (v/v) aqueous acetonitrile), spotted onto a target plate, and dried under vacuum. For the MS/MS studies, the collision energy was set to 1 kV, and argon was used as collision gas. The 4700 Calibration standard kit, calmix (Applied Biosystems), was used as the external calibrant for the MS mode, and [Glu1] fibrinopeptide B human (Sigma) was used as an external calibrant for the MS/MS mode.

For analysis of mass spectra, data were processed using Data Explorer 4.9 Software (Applied Biosystems). The processed spectra were subjected to manual assignment and annotation with the aid of a glycobioinformatics tool, GlycoWorkBench [39]. The proposed assignments for the selected peaks were based on $^{12}$C isotopic composition together with knowledge of the bio-synthetic pathways. Proposed structures were then confirmed by data obtained from MS/MS.

## Statistical analysis

All data acquired from the MS analysis were processed according to previous work [40]. For a given spectrum, data were normalised by dividing the relative intensity of an identified peak, corresponding to a N-glycan, to the sum of the relative intensities of all identified peaks. The total relative abundance for each structural feature (variable, i.e. high mannose, mono-, bi-, tri- and tetra-antennary N-glycans, agalactosylated, core-fucosylated and bisected N-glycans, and LacNAc, NeuAc, Lewis$^X$ terminal epitopes) was calculated by summing the normalised relative intensity of each identified N-glycan that contained the corresponding structural fea-ture. Specifically, the total relative abundance of terminal epitopes (LacNAc, NeuAc, Lewis$^X$) was calculated taking also into account the number of terminal epitopes on each identified peak. The total relative abundance of the antennae (mono-, bi-, tri- and tetra-antennary N-gly-cans) and of the terminal epitopes (LacNAc, NeuAc, Lewis$^X$) was expressed as percentage.

All statistical analyses were performed using SPSS version 25. A paired $t$-test was applied to analyse differences for paired hCG samples (EP-hCG versus LP-hCG), while an independent sample $t$-test was used for differences between EP-hCG and GTD-hCG. Significance was taken as $P < 0.05$. The significance threshold was corrected for multiple comparisons using Bonfer-roni method. For the paired $t$-test epitopes variables (LacNAc, NeuAc and Lewis$^X$), the cor-rected $P$ was set at <0.0166, while for the N-glycans variable (high mannose, mono-, bi-, tri- and tetra-antennary N-glycans, agalactosylated, bisected and core-fucosylated N-glycans) it was set at <0.0063. For the independent sample $t$-test epitopes variables (LacNAc, NeuAc and Lewis$^X$), the corrected $P$ was set at <0.0166, while for the N-glycans variable (mono-, tri- and tetra-antennary N-glycans, agalactosylated, bisected and core-fucosylated N-glycans) it was set at <0.0083. The Bejamini and Hochberg method for false discovery rate was also applied. The method was available in the SPSS software package through the syntax command following instructions published on the IBM support web page (Bejamini and Hochberg, document number 418001). The strictest result of the above methods was taken into account in terms of corrected significance. For the paired $t$-test, the test of normality with Shapiro-Wilk's statistic and outliers was taken into consideration for the values resulting from the differences of the above variables (EP-hCG versus LP-hCG). For the independent sample $t$-test, the test of nor-mality with Shapiro-Wilk's statistic, outliers, and variance according to Levene's test for equal-ity of variances were taken into consideration. Variables not meeting the above assumptions were excluded from the analysis.

Hierarchical cluster analysis was applied using between-groups linkage as a clustering method and Euclidean distance as a measure interval (similar results were obtained with other clustering methods and measure intervals). On EP-hCG versus GTD-hCG samples cluster

**Table 1. hCG concentration from individual samples of EP-hCG and LP-hCG.**

| Patient Id. (early/late pregnancy) | Gestational age of sample (weeks+days) | Sample [hCG] (IU/L) | [hCG]* after purification (mg/L) |
|---|---|---|---|
| EP-hCG1 | 9+1 | 835584 | 1022 |
| LP-hCG1 | 20+1 | 75568 | 221 |
| EP-hCG2 | 7+4 | 369648 | 257 |
| LP-hCG2 | 20+6 | 63107 | 52 |
| EP-hCG3 | 7+1 | 120101 | 138 |
| LP-hCG3 | 20+1 | 11468 | 17 |
| EP-hCG4** | 7+6 | 183382 | 389 |
| LP-hCG4** | 20+3 | 96163 | 89 |

Details of paired EP-hCG and LP-hCG urine samples collected from healthy pregnant women. Number following "hCG" corresponds to patient number.

* [hCG] denotes hCG concentration.

** Samples identified as pre-eclampsia samples and were excluded from further analysis.

analysis was performed for the variables found to be significant after correction for multiple comparisons using the criteria as stated above (core fucosylation, bisected, LacNAc, Lewis$^X$, NeuAc). For the complete set of samples (EP-hCG, LP-hCG and GTD-hCG) hierarchical cluster analysis was performed using as variables the NeuAc and bisected N-glycans.

## Results

Glycomics analysis of N-glycans was carried out on a total of 12 hCG samples isolated from the urine of 4 pregnant as well as 4 women with GTD-hCG. Of the 4 samples collected from patients with normal pregnancy, one pair (EP-hCG4 and LP-hCG4) was excluded from the statistical analysis because the patient was subsequently identified as having late onset pre-eclampsia. Details including gestational age of collected urine sample, hCG concentration and concentration following chromatographic purification for samples derived from normal pregnancy are given in **Table 1**. In addition the histological diagnosis and clinical outcome (benign vs malignant) for samples collected from patients with a molar pregnancy are given in **Table 2**. Of the 4 samples collected from patients suffering molar pregnancy, 3 were benign with regressing serum hCG levels following molar evacuation, while one sample (GTD-hCG14) had persistent trophoblastic neoplasia requiring multiagent chemotherapy to achieve normalisation of serum hCG and disease resolution.

The N-glycan profiles of all hCG samples derived from the above pregnancy conditions were determined using a glycomics strategy previously optimised for the characterisation of N-glycomes from a wide variety of samples including pregnancy associated glycoproteins,

**Table 2. hCG concentration from individual samples of GTD-hCG.**

| Patient Id. (Molar pregnancy) | Time from evacuation (days) | Sample hCG (IU/L) | [hCG] after purification (mg/L) | Histology (Mole type) | Benign or Malignant? |
|---|---|---|---|---|---|
| GTD-hCG1 | 0 | 693492 | 1297 | CHM | Benign |
| GTD-hCG2 | 0 | 1276646 | 2978 | CHM | Benign |
| GTD-hCG3 | 0 | 64448 | 361 | CHM | Benign |
| GTD-hCG4* | 0 | 7704768 | 11544 | CHM | Malignant |

Details of urine samples collected from patients suffering molar pregnancy. All molar pregnancies were subsequently histologically diagnosed as complete moles (CHM).

* Patient identity GTD-hCG4 had persistent trophoblastic neoplasia requiring multiagent chemotherapy to achieve normalisation of serum hCG and disease resolution.

such as human glycodelin [41]. This glycomic strategy is based on MALDI-MS and MS/MS analyses of the total population of N-glycans after their release from the polypeptide backbone of hCG via digestion with peptide N-glycosidase F. The glycans were permethylated prior to MALDI analysis in order to enhance sensitivity and to allow unambiguous MS/MS fragmentation. The MALDI-MS experiments defined glycan compositions whilst MS/MS experiments determined antennae sequences, as well as the location of fucose residues in Lewis structures. Evidence for bisected glycans was obtained from observing whether glycans which had been shown by MS/MS to contain terminal GlcNAc residues were capped with a galactose residue after treatment with β-galactosyltransferase and UDP-Gal. Bisected GlcNAc residues are not accessible to β-galactosyltransferase and therefore are not capped with a galactose residue during this procedure. In contrast, terminal GlcNAc residues on truncated N-glycan antennae are fully accessible to the β-galactosyltransferase. Therefore, their capping with galactose serves as a positive control for β-galactosyltransferase activity. This methodology has been previously shown to be a reliable way of confirming bisected GlcNAc [38]. A representative MALDI-TOF MS profile of a sample from each group is depicted in **Fig 1** (MALDI profiles of other hCG samples are shown in figures in **S1** and **S2 Figs**). All EP-hCG and LP-hCG samples were characterised by high mannose (*m/z* 1579, 1783), mono- (*m/z* 1620, 1982), bi- (*m/z* 2244, 2431, 2489, 2605), tri- (*m/z* 2693, 2938, 3112, 3286) and tetra-antennary (*m/z* 3142, 3561, 3735, 3923) complex-type N-glycans all mainly core fucosylated (**Fig 1A and 1B** and **Table 3**). On those hCG samples, N-glycans were predominantly bi-antennary (mean values of 56.7 and 62.6% respectively) and tri-antennary structures (21.8 and 24.9% respectively), while tetra-antennary structures were of minor abundance (3.8 and 3.5% respectively). The main capping sugars on their antennae corresponded to undecorated *N*-acetyllactosamine (LacNAc) units (62.4 and 67.7% respectively) followed by NeuAc residues (27.2 and 17.6% respectively) and Lewis[X] epitopes (10.4 and 14.7% respectively, figure in **S3 Fig**), the latter being verified by the elimination of a fucose residue from the corresponding molecular ion. A β-galactosyltransferase experiment (shown in figure in **S4 Fig**) indicated the presence of bisected N-glycans ranging from bi- to tetra-antennary structures (**Fig 1A and 1B** red peaks). Calculation of the relative intensities of the bisected N-glycans showed that their mean value shifted from 23.6% for the EP-hCG samples to 44.9% for the LP-hCG samples (**Table 3**). Of note is the fact that all EP-hCG and LP-hCG samples were characterised by the presence of high abundance pauci-mannose type N-glycans (see *m/z* 1141 and 1345,figure in **S5 Fig**).

On the contrary, GTD-hCG samples contained fewer glycoforms compared to those seen in EP-hCG and LP-hCG (**Fig 1C** and **figure in S2 Fig**). GTD-hCG samples were characterised by the minor abundance of high mannose N-glycans (mean value 2.2%) and by the presence of complex N-glycans of which the most abundant were mono- and bi-antennary structures (22.3% and 66.0% respectively, **Table 3**). The latter structures were predominantly capped by NeuAc residues (79.4%), while other non-reducing terminal epitopes were of minor abundance (18.8 and 1.8% for LacNAc and Lewis[X] respectively). In contrast to the EP-hCG and LP-hCG samples, the paucimannose type N-glycans were not found consistently in high abundance in the GTD-hCG samples (see *m/z* 1141 and 1345, figure in **S5 Fig**). Pre-eclampsia samples (EP-hCG4 and LP-hCG4) exhibited N-glycomic profiles consisting of structures found in both the EP/LP-hCG and GTD-hCG samples. They were characterised by the presence of bisected bi- and tri-antennary N-glycans and Lewis[X] terminal epitopes (as of EP/LP-hCG), but also by the high abundance of sialylated bi-antennary N-glycans (as of GTD-hCG, *m/z* 2792 and 2966, figure in **S6 Fig**).

Statistical analysis was applied on the hCG samples (see tables in **S3–S6 Tables**). All structural features that were identified from the MALDI-TOF MS analysis have been used as variables by calculating the total relative intensity of each structural feature for each sample

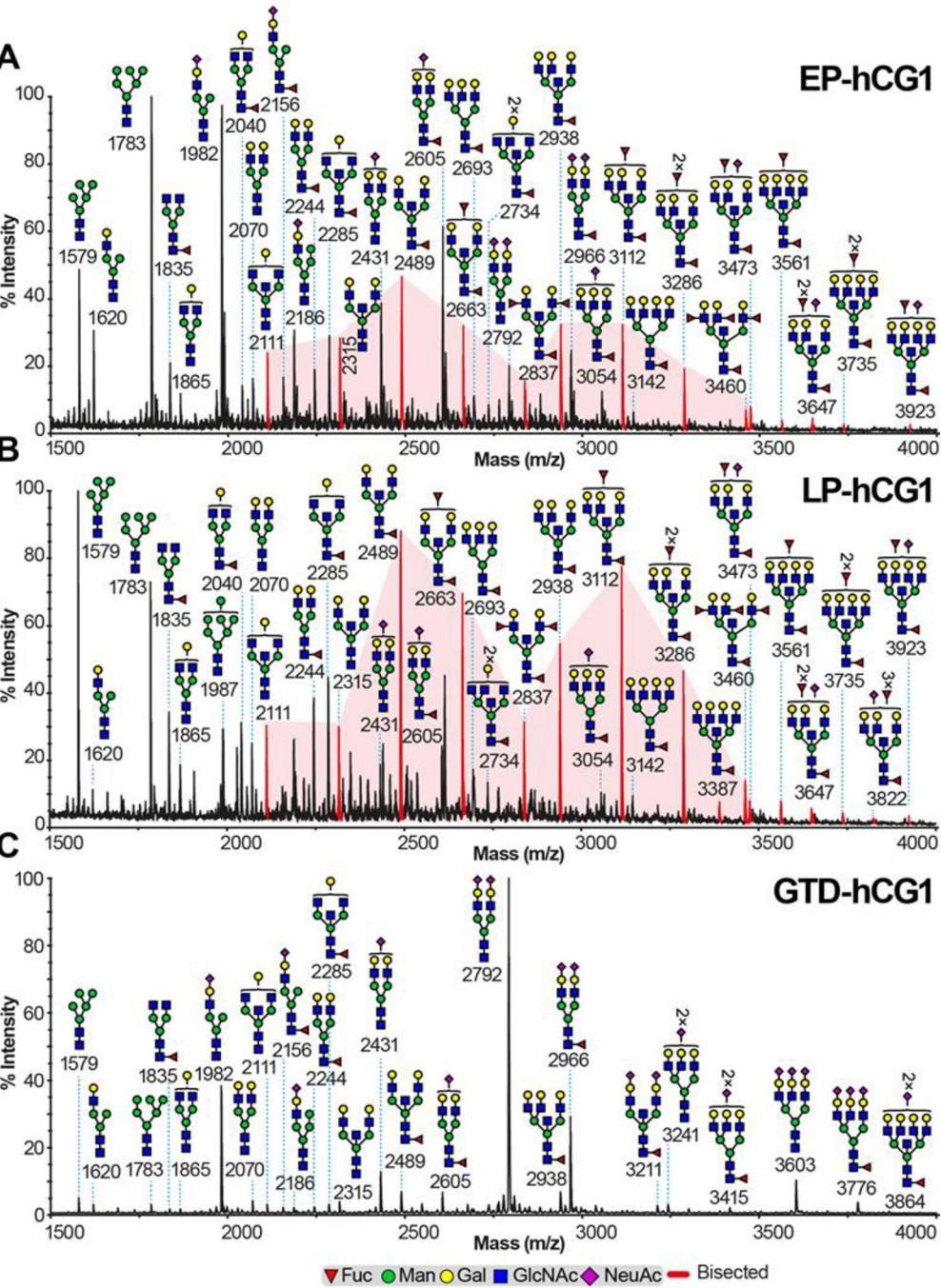

**Fig 1. N-glycomic profiles of hCG samples from normal pregnancy and GTD.** MALDI-TOF MS spectra of permethylated N-glycans derived from (**A**) EP-hCG1, (**B**) LP-hCG1 and (**C**) GTD-hCG1 samples. Structures above a bracket were not unequivocally defined. Red peaks correspond to bisected N-glycans with various antenna configurations. Putative structures are based on composition, tandem MS, β-galactosyltransferase experiment and knowledge of biosynthetic pathways. All molecular ions are [M+Na]$^{+}$. MALDI-TOF MS spectra of other hCG samples are shown in **S1 Fig**.

(**Table 3**). Paired *t*-test on EP-hCG vs LP-hCG samples showed that among all the aforementioned structural features only bisected N-glycans were statistically significant at the $P < 0.05$ level (**Table 3** and **Fig 2**). The results of the independent *t*-test on EP-hCG versus GTD-hCG

**Table 3. Comparison of the structural features (variables) used for the analysis of hCG samples.**

| Structural feature/Variable | EP-hCG mean±SEM | LP-hCG mean±SEM | Paired *t*-test (1) | GTD-hCG mean±SEM | Ind. *t*-test (2) |
|---|---|---|---|---|---|
| High mannose | 40.4±21.1 | 28.5±9.4 | 0.417 | 2.2±0.9 | (3) |
| Mono-antennary | 17.7±5.0 | 9.0±0.3 | 0.238 | 22.3±2.4 | 0.402 |
| Bi-antennary | 56.7±3.0 | 62.6±1.2 | 0.113 | 66.0±1.6 | (3) |
| Tri-antennary | 21.8±1.9 | 24.9±0.9 | 0.299 | 10.9±2.0 | 0.013 |
| Tetra-antennary | 3.8±1.1 | 3.5±0.5 | 0.766 | 0.7±0.4 | 0.030 |
| Agalactosylated | 24.7±6.6 | 30.0±1.7 | 0.401 | 5.8±2.1 | 0.092 |
| Bisected | 23.6±1.4 | 44.9±3.1 | **0.006** | 6.4±2.0 | **0.001** |
| Core-fucosylated | 60.4±5.6 | 66.7±2.2 | 0.473 | 34.1±1.5 | **0.003** |
| LacNAc | 62.4±4.0 | 67.7±1.0 | 0.344 | 18.8±4.6 | **0.001** |
| NeuAc | 27.2±2.8 | 17.6±2.9 | 0.232 | 79.4±4.9 | **0.001** |
| Lewis[X] | 10.4±2.1 | 14.7±2.5 | 0.131 | 1.8±.0.4 | **0.005** |

Mean values ± standard error of the mean (SEM) of the structural features (variables) used for the analysis of hCG samples, *t*-test for paired EP-hCG vs LP-hCG samples, and *t*-test for independent EP-hCG vs GTD-hCG samples (full data, see tables in **S3–S6 Tables**). Values in bold correspond to variables significant after correction for multiple comparison according to Bonferroni test

(1) Paired *t*-test of EP-hCG vs LP-hCG samples.

(2) Independent *t*-test of EP-hCG vs GTD-hCG samples.

(3) High mannose and bi-antennary N-glycans did not meet normality test in the Independent *t*-test.

samples showed that bisected and core-fucosylated N-glycans were statistically significant at least at the $P < 0.05$ level. Terminal epitopes such as LacNAc and Lewis[X] were also found significant with the most profound difference detected being the increased abundance of NeuAc epitopes at the $P < 0.05$ level.

The identified N-glycan structural features could potentially be used as biomarkers. Therefore, we sought to investigate whether the above N-glycans could be used to differentiate pregnancy conditions. To this end, the EP-hCG and GTD-hCG samples were subjected to hierarchical cluster analysis (HCA) using as clustering variables the structural features with statistical significance at the $P < 0.05$ level (bisected and core-fucosylated N-glycans, and LacNAc, Lewis[X] and NeuAc epitopes, **Fig 2** and **Table 3**). HCA successfully grouped the EP and GTD samples into two distinct clusters (**Fig 3A**). HCA was also applied in order to differentiate all discussed pregnancy conditions using as clustering variables the bisected N-glycans and the NeuAc epitope (as the most significant variables). HCA successfully grouped the EP, LP and GTD samples in three distinct clusters using as variables only the bisected N-glycans and the NeuAc epitope (**Fig 3B**).

## Discussion

In this study, we analysed the N-glycans expressed on hCG isolated from the urine of four women during early (EP-hCG) and late (LP-hCG) pregnancy as well as hCG from four women who had been diagnosed with GTD. One woman (EP-hCG4 and LP-hCG4) was excluded from our statistical analysis because she was identified as having late onset pre-eclampsia. Although statistical analysis on low numbers of samples is not ideal, it does provide an informative guide to the structural features that characterise the different hCG preparations. The main objective of this study was rigorous structural identification of N-glycans which had not previously been identified on hCG.

The three healthy EP- samples shared the same glycan repertoires, albeit with considerable inter-individual variation in relative abundances. The pregnancy hCG glycome was found to

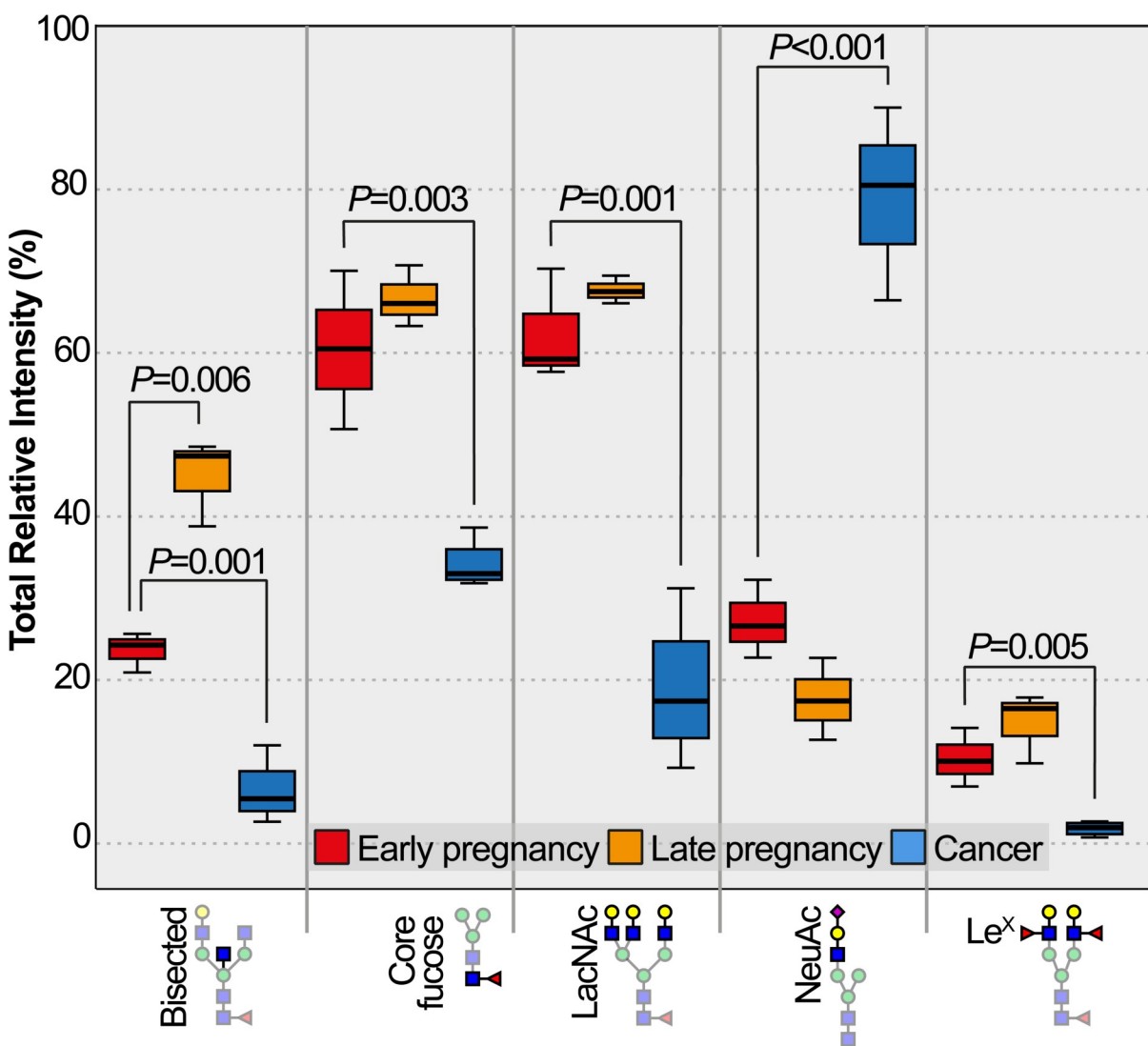

**Fig 2. Structural features of N-glycans on all hCG samples found significantly different.** For EP-hCG vs LP-hCG samples the only significant structural feature found was the bisected N-glycans. For EP-hCG vs GTD-hCG samples the significant structural features found were the bisected and core-fucosylated N-glycans, and the LacNAc, NeuAc and Lewis[x] terminal epitopes. For EP-hCG vs LP-hCG samples, paired *t*-test; for EP-hCG vs GTD-hCG samples, independent *t*-test. *P* values were corrected according to Bonferroni method.

be substantially more diverse than the GTD-hCG glycans. The latter are almost exclusively detected as sialylated structures ($P = 0.001$ which is consistent with previous literature [1, 29]. **Fig 4** summarises the N-glycan structures that are newly identified in this study. EP and LP hCG N-glycans contain bisected N-glycans in various antenna configurations (from bi- to tetra-antennary), none of which has previously been identified on hCG. Interestingly, bisected type N-glycans was the only significant discriminating variable between EP-hCG and LP-hCG ($P = 0.006$). Moreover, a certain number of them were found to be capped with Lewis[x] antigens, instead of the terminal sialylation which is characteristic of the non-bisected glycans. This observation contradicts the prevailing view that hyperglycosylation in early pregnancy is associated with higher abundance of multi-antennary N-glycans compared with later in pregnancy [15, 29]. Lastly, the presence of high mannose and terminal Lewis[x] may play a role in immunosuppression via binding to lectins such as DC-SIGN, but this remains to be investigated.

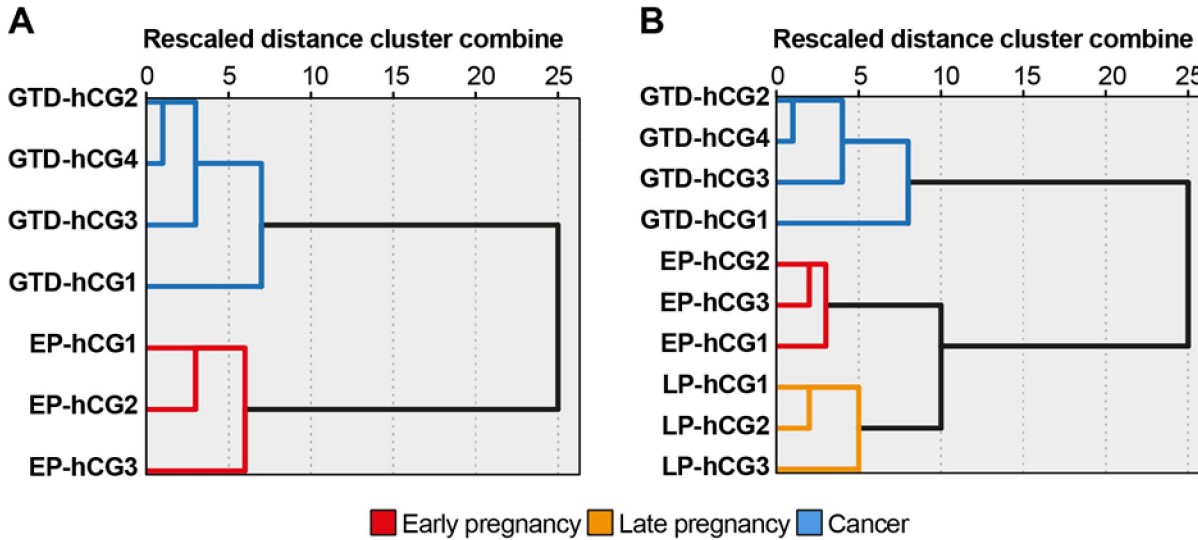

**Fig 3. Hierarchical cluster analysis of hCG samples.** (**A**) Hierarchical cluster analysis of EP-hCG and GTD-hCG samples using as clustering variables the total relative abundance of the structural features that were significant at the $P < 0.05$ level (after correction for multiple comparison with Bonferroni test) as found from independent $t$-test and shown on **Fig 2**. **The variables** bisected and core-fucosylated N-glycans, and LacNAc, NeuAc and Lewis[X] terminal epitopes. (**B**) Hierarchical cluster analysis of all hCG samples (EP, LP and GTD) using as clustering variables the total relative abundance of bisected N-glycans and NeuAc terminal epitopes. For hierarchical cluster methods, see materials and methods.

Interestingly, EP-hCG4 and LP-hCG4, known to have late onset pre-eclampsia, shared properties of the healthy normal pregnancy and GTD pregnancy hCG. In particular, we noted the presence of bisected bi- and tri-antennary N-glycans and Lewis[X] terminal epitopes and an increase in the bisected structures as we move from EP to LP (as of EP/LP-hCG). In addition, we noted a higher abundance of sialylated bi-antennary N-glycans compared to normal EP-hCG and LP-hCG, similar to what we saw with GTD-hCG. Pregnancy can be complicated by many different disorders, such as miscarriage, pre-eclampsia and gestational diabetes. Detecting differential glycosylation patterns of hCG may allow us to accurately screen for these complications during early pregnancy. Obtaining early and accurate diagnosis of these complications could help decrease the risk to the fetus and mother by enabling health care providers to diagnose, treat and manage conditions before they become serious.

It was reported over twenty years ago that surface expression of bisected-type N-glycans inhibits natural killer (NK) cell-mediated lysis [42]. NK cells represent about 70% of the lymphocytes in the endometrium prior to implantation [43]. Therefore, the presence of bisected type N-glycans in pregnancy hCG is consistent with a role for this hormone during the initial invasion of the endometrial lining by the early human embryo. Placental trophoblasts secrete monocyte inflammatory protein-1α (MIP-1α) that induces the chemotaxis of NK cells and monocytes to their surface [44]. It is known that hCG derived from pregnancy urine inhibits NK cell cytotoxicity *in vitro* at nanomolar concentrations [45]. This inhibitory activity was attributed to the hormonal influence of hCG. The results of the current glycomic analysis suggest the possibility that hCG employs its bisected type glycans as functional groups to suppress NK cytotoxicity *in vivo*. However, this assignment must be experimentally confirmed. Trophoblasts lack human leukocyte antigen (HLA) class I expression, which could potentially make them more susceptible to NK cell-mediated cytotoxicity [46, 47]. However, HLA class I negative cell types can evade lysis by NK cells if they express sufficient amounts of bisected type N-glycans [42, 48]. Recent glycomics analysis of human trophoblasts indicate that they express a

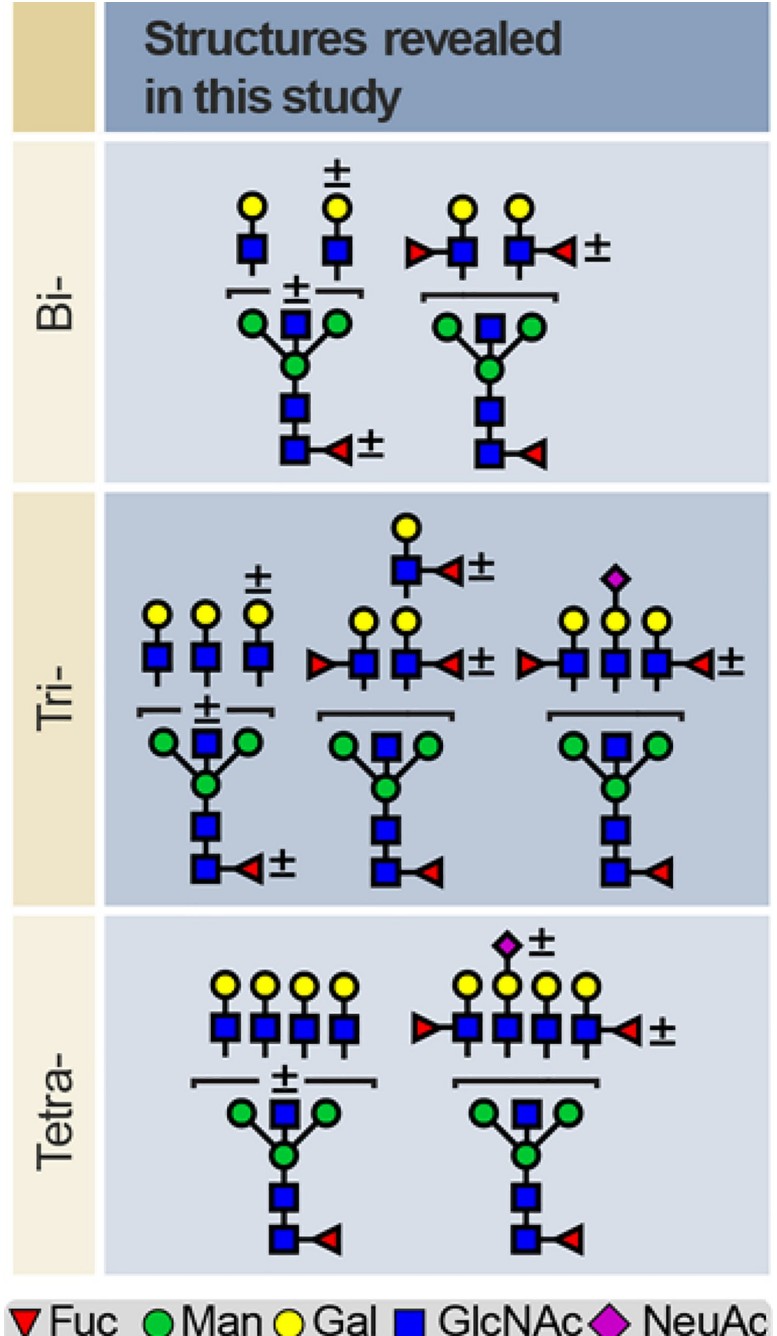

**Fig 4. Major new N-glycan structures found in this study.** Bi-, tri- and tetra-antennary N-glycans detected in early and late pregnancy hCG samples. Plus, or minus sign (±) above or to the right of each residue correspond to the presence or absence of the corresponding residue. Note that fucosylated N-glycans are detected mainly as bisected structures.

much higher level of bisected type glycans compared to HLA class I positive cells and tissues, consistent with their ability to resist NK cell cytotoxicity [49].

Bisected N-glycans terminated with Lewis[X] antigens are rare in humans. Interestingly, apart from pregnancy hCG described herein, sperm are the only reported human source of

substantial amounts of this family of glycans, which have been implicated in the inhibition of immune responses [50, 51]. There have been many models presented to explain how the pregnant human female accommodates the foreign fetus during pregnancy. One major factor that is often overlooked is glycosylation. There are now many soluble and cell surface associated glycoproteins that have been implicated in this protective effect, including glycodelin-A and CA125 [52]. A model for the induction of tolerance mediated primarily by carbohydrate recognition is known as the human fetoembryonic defence system (hu-FEDS) hypothesis [52, 53]. hCG could soon be added to this group of glycoproteins in this model if evidence can be obtained that its glycans act as functional groups to suppress NK cell cytotoxicity. The Hu-FEDS hypothesis also predicts that tumour cells employ the same carbohydrate functional groups used in the pregnant uterus to evade the human immune response [52]. Consistent with this concept, hCG and other pregnancy-related glycoproteins are often upregulated in tumour cells [54–57].

Compared to the EP- and LP-hCG glycome, the GTD-hCG structures exhibited a significant change in various structural features, such as antenna configuration and terminal epitopes, the most characteristic being the increase of NeuAc residues. An immunomodulatory role of sialic acid in pregnancy has previously been suggested. Thus, sialylated glycoconjugates have been shown to modulate the host dendritic cell function by suppressing the production of the pro-inflammatory cytokine IL-12, as well as T cell activation and proliferation in response to mitogens and antigens [58]. Changes in the activity of sialic acid regulatory enzymes such as sialyltransferases and sialidase have been associated with diseased states in pregnancy such as gestational diabetes [41]. High expression of sialyltransferase have been reported in different cancers and shown to correlate with increase motility, invasion, and metastatic potential of tumour cells [59]. It is therefore interesting to observe the change in sialylation in our GTD samples and further question the significance of sialylation in hCG. Of particular note is the high abundance of sialylated glycoforms at both early and late pregnancy in the sample complicated by pre-eclampsia (see supplementary data). This hCG was obtained from a woman who developed late onset pre-eclampsia but delivered at term. It is interesting that her EP and LP hCG profiles are dominated by the sialylated glycans that are abundant in GTD. It is conceivable that high levels of hCG sialylation continuing through pregnancy might be a risk factor for the development of pre-eclampsia and possibly other obstetric syndromes.

Hyperglycosylated hCG is said to be larger than hCG because of additional sugar residues [29]. EP-hCG has been reported as hyperglycosylated [31]. Although we did not find higher levels of multiantennary N-glycans in EP- compared with LP-hCG, our results showed an increase in sialylated structures in EP-hCG and GTD-hCG, compared to LP-hCG. It is known that sialic acid containing glycoproteins can run aberrantly on SDS-PAGE gels [60]. Therefore, our data suggest that the hCG-h bands detected by the B152 antibody could be highly sialylated glycoforms that share similar levels of N-glycan branching observed in EP-hCG and GTD-hCG.

In conclusion, by employing glycomic technologies we have provided new insights into hCG N-glycosylation during pregnancy. We have identified novel glycan epitopes which potentially constitute new targets for diagnostics and for future investigations into the roles of glycans in hCG function and in the preservation of a healthy pregnancy.

## Supporting information

**S1 Table. NanoLC-MS/MS analysis of hCG-3 GTD11.** Raw data peak picking was performed using Mascot.dll v1.6.0.13 (Applied Biosystems) and Mascot (version 2.6, www.matrixscience.com) was used to search the NCBIProt database for sequences consistent with the MS/MS

fragment ions. Peak list generation and database searching were conducted with the default parameters. MS/MS data were used to search the *Homo sapiens* portion of the NCBIProt protein database with the following parameters: monoisotopic peptide masses, allowing for partial oxidation of methionine residues and carboxymethylation of cysteine residues, mass tolerance of 300 ppm. and fragment ion tolerance of 300 ppm. Tryptic digests of up to 1 missed cleavage were tolerated. Proteins listed contain 3 or more peptides with ion scores higher than 38, which indicates identity or extensive sequence similarity (p<0.05).
(PDF)

**S2 Table. NanoLC-MS/MS analysis of hCG-4 GTD10a.** Raw data peak picking was performed using Mascot.dll v1.6.0.13 (Applied Biosystems) and Mascot (version 2.6, www. matrixscience.com) was used to search the NCBIProt database for sequences consistent with the MS/MS fragment ions. Peak list generation and database searching were conducted with the default parameters. MS/MS data were used to search the Homo sapiens portion of the NCBIProt protein database with the following parameters: monoisotopic peptide masses, allowing for partial oxidation of methionine residues and carboxymethylation of cysteine residues, mass tolerance of 300 ppm. and fragment ion tolerance of 300 ppm. Tryptic digests of up to 1 missed cleavage were tolerated. Proteins listed contain 3 or more peptides with ion scores higher than 38, which indicates identity or extensive sequence similarity (p<0.05).
(PDF)

**S3 Table. Normal distribution for paired samples *t*-test.** Normal distribution test for the difference (Δ) between early (EP-hCG) and late (LP-hCG) pregnancy variables (structural features) using the Shapiro-Wilk statistic.
(PDF)

**S4 Table. Paired samples *t*-test analysis for EP-hCG and LP-hCG samples.** Variables from EP-hCG (E) and LP-hCG samples (L) meeting normal distribution (**S1 Table**) were subjected to paired samples *t*-test. Value in bold correspond to variable significant at *P*<0.05.
(PDF)

**S5 Table. Normal distribution test for independent samples *t*-test.** Variables (structural features) deriving from the EP-hCG and GTD-hCG were subjected to the Shapiro-Wilk statistic. Values in bold correspond to variables significant at *P*<0.05 (not meeting normal distribution assumption). Note that high mannose (**HM**) and bi-antennary N-glycans (**Bi-**) did not meet the normal distribution assumption and therefore they were excluded from further analysis.
(PDF)

**S6 Table. Independent samples *t*-test analysis for EP-hCG and GTD-hCG samples.** Variables from EP-hCG and GTD-hCG samples meeting normal distribution (**S3 Table**) were subjected to independent samples *t*-test. Note that mono-antennary (**Mono-**) and agalactosylated (**Agalacto**) N-glycans were not found statistically significant (*P* = 0.402 and *P* = 0.092 respectively). Values in bold correspond to variables significant at *P*<0.05 for the equality of variances (var.) and equality of means.
(PDF)

**S1 Fig. N-glycomic profiles of hCG samples.** MALDI-TOF MS spectra of permethylated N-glycans derived from (**A**) EP-hCG2, (**B**) LP-hCG2, (**C**) EP-hCG3 and (**D**) LP-hCG3 samples. Structures above a bracket were not unequivocally defined. Red peaks correspond to bisected N-glycans with various antenna configurations. (**C, D**) Horizontal line above the spectra indicates the area and the level of zoom. Putative structures are based on composition, tandem

MS, β-galactosyltransferase experiment and knowledge of biosynthetic pathways. All molecular ions are [M+Na]$^{+}$.
(PDF)

**S2 Fig. N-glycomic profiles of GTD-hCG samples.** MALDI-TOF MS spectra of permethylated N-glycans derived from (**A**) GTD-hCG2, (**B**) GTD-hCG3 and (**C**) GTD-hCG4 samples. Structures above a bracket were not unequivocally defined. Putative structures are based on composition, tandem MS. All molecular ions are [M+Na]$^{+}$.
(PDF)

**S3 Fig. EP and LP hCG samples contain Lewis$^{X}$ antigens.** Representative MALDI-TOF/TOF MS/MS spectra of the molecular ions found at (**A, B**) *m/z* 3286 and (**C, D**) 3460 for the EP and LP-hCG1 samples respectively. Horizontal dashed lines correspond to indicated losses from the molecular ion [M+Na]$^{+}$. For (A and B), the fragment ion at *m/z* 3080 corresponds to the elimination of a fucose residue from the molecular ion at *m/z* 3286, indicative of a fucose residue being on the C3 of GlcNAc residue (Lewis-X). Similarly, for (C and D), the fragment ion at *m/z* 3254, corresponds to the elimination of a fucose residue from the molecular ion at *m/z* 3460.
(PDF)

**S4 Fig. β-Galactosyltransferase (GalT) experiment on hCG1 samples.** MALDI-TOF MS spectra of permethylated N-glycans derived from (**A**) EP-hCG1 sample (top panel, control, same as Fig 1A; lower panel, after GalT) and (**B**) LP-hCG1 sample (top panel, control, same as Fig 1B; lower panel, after GalT). Note the molecular ions at *m/z* 2315, 2489, 2663, 2837, 2938, 3112, 3286 3460, 3473 and 3647 on EP-hCG1 and LP-hCG1 spectra before the GalT experiment (**A** and **B**, top panels), which correspond to complex mature N-glycans with a single terminal GlcNAc residue. The same ions were also detected on the EP-hCG1 and LP-hCG1 spectra after the GalT experiment (**A** and **B**, lower panels), indicating that those terminal GlcNAc residues are not subject to a GalT experiment. Therefore, the above molecular ions correspond to bisected N-glycan structures. On the contrary, on EP-hCG1 and LP-hCG1 spectra before the GalT experiment (**A** and **B**, top panels), the molecular ions at *m/z* 1835 and 1865 correspond to agalactosylated N-glycans (GlcNAc terminated residues on mannose arms). These ions are not present in GalT samples (**A** and **B** lower panels) indicating that these ions are subject to the β-galactosyltransferase experiment further indicating that they do not correspond to bisected N-glycans. Putative structures are based on composition, tandem MS and knowledge of biosynthetic pathways. All molecular ions are [M+Na]$^{+}$.
(PDF)

**S5 Fig. Paucimannose type N-glycans on hCG samples.** Partial MALDI-TOF MS spectra of permethylated N-glycans derived from EP-hCG, LP-hCG and GTD-hCG samples as indicated on the corresponding spectra. Red peaks correspond to paucimannose type N-glycans. For comparison of the relative intensity of the paucimannose type N-glycans compared to the relative intensity of molecular ions as shown in Fig 1 and S1 Fig, compare the ratio of paucimannose type N-glycans to the high mannose N-glycans found at *m/z* 1579 and 1783. All molecular ions are [M+Na]$^{+}$.
(PDF)

**S6 Fig. N-glycomic profiles of pre-eclampsia samples.** MALDI-TOF MS spectra of permethylated N-glycans derived from (**A**) EP-hCG4 and (**B**) LP-hCG4 samples. Putative structures are based on composition, tandem MS. All molecular ions are [M+Na]$^{+}$.
(PDF)

## Acknowledgments

We thank Wilson Stuart at Tayside NHS, Scotland, UK for support with purification of hCG.

## Author Contributions

**Conceptualization:** Linda Ibeto, Paola Grassi, Poh-Choo Pang, Julian Norman Taylor, Paula Almeida, Richard Harvey, Michael Seckl, Stuart M. Haslam, Anne Dell.

**Data curation:** Linda Ibeto, Paola Grassi, Shabnam Bobdiwala, Maya Al-Memar, Julian Norman Taylor, Paula Almeida, Richard Harvey, Tom Bourne, Michael Seckl.

**Formal analysis:** Linda Ibeto, Aristotelis Antonopoulos, Paola Grassi, Poh-Choo Pang, Maria Panico, Mark R. Johnson, Gary Clark, Stuart M. Haslam, Anne Dell.

**Funding acquisition:** Paul Davis, Mark Davis.

**Investigation:** Linda Ibeto, Stuart M. Haslam.

**Methodology:** Linda Ibeto, Aristotelis Antonopoulos, Paola Grassi, Tom Bourne, Stuart M. Haslam, Anne Dell.

**Project administration:** Anne Dell.

**Software:** Linda Ibeto.

**Supervision:** Julian Norman Taylor, Paula Almeida, Mark R. Johnson, Gary Clark, Stuart M. Haslam, Anne Dell.

**Writing – original draft:** Linda Ibeto, Mark R. Johnson, Richard Harvey, Michael Seckl, Gary Clark, Stuart M. Haslam, Anne Dell.

**Writing – review & editing:** Linda Ibeto, Aristotelis Antonopoulos, Paola Grassi, Poh-Choo Pang, Maria Panico, Shabnam Bobdiwala, Maya Al-Memar, Julian Norman Taylor, Paula Almeida, Mark R. Johnson, Richard Harvey, Tom Bourne, Michael Seckl, Gary Clark, Stuart M. Haslam, Anne Dell.

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
