## [Decision Letter · Decision Letter 0]

14 Oct 2019

PONE-D-19-25594

Insights into the hyperglycosylation of human chorionic gonadotropin revealed by glycomics analysis

PLOS ONE

Dear Dr ibeto,

Thank you for submitting your manuscript to PLOS ONE. After careful consideration, we feel that it has merit but does not fully meet PLOS ONE’s publication criteria as it currently stands. Therefore, we invite you to submit a revised version of the manuscript that addresses the points raised during the review process. Both reviewers raised constructive criticisms to methodological aspects of the present work. These issues need to be carefully addressed before submission. We understand that novel experiments may be necessary for the completion of the revised version of the manuscript.

We would appreciate receiving your revised manuscript by Nov 28 2019 11:59PM. To enhance the reproducibility of your results, we recommend that if applicable you deposit your laboratory protocols in protocols.io, where a protocol can be assigned its own identifier (DOI) such that it can be cited independently in the future. For instructions see: http://journals.plos.org/plosone/s/submission-guidelines#loc-laboratory-protocols

We look forward to receiving your revised manuscript.

Kind regards,

Roger Chammas, M.D, Ph.D

Academic Editor

PLOS ONE

Journal Requirements:

2. In the ethics statement in the manuscript and in the online submission form, please provide additional information about the urine samples used in your retrospective study, including: a) whether all samples were fully anonymized before you accessed them and b) the date range (month and year) during which patients' urine samples were accessed.

The author(s) received no specific funding for this work

We note that one or more of the authors are employed by a commercial company: Mologic LTD, Bedford Technology Park, Bedfordshire, UK.

Reviewers' comments:

Reviewer's Responses to Questions

**Comments to the Author**

1. Is the manuscript technically sound, and do the data support the conclusions?

Reviewer #1: Yes

Reviewer #2: Yes

2. Has the statistical analysis been performed appropriately and rigorously? 

Reviewer #1: Yes

Reviewer #2: Yes

3. Have the authors made all data underlying the findings in their manuscript fully available?

Reviewer #1: Yes

Reviewer #2: No

4. Is the manuscript presented in an intelligible fashion and written in standard English?

Reviewer #1: Yes

Reviewer #2: Yes

5. Review Comments to the Author

Reviewer #1: profile among early and late stage pregnancy and gestational trophoblastic disease. The following points should be addressed.

1. It would be very useful to show the measured proteomes of representative hCG samples. This would demonstrate the extent to which observed released glycans can be attributed to hCG, rather than co-isolated glycoproteins.

2. Please discuss the rationale for using MALDI TOF MS profiling of glycans. LC-MS has better dynamic range and would be the logical choice for quantitative studies.

3. How is “ultrasensitive” MALDI distinguished from traditional MALDI?

4. It does not appear that the authors released O-glycans. Doing so would have allowed them to shed light on the extent to which hyperglycosylation of hCG is due to changes in O-glycosylation as mentioned on p. 3, line 74.

5. The quantitative interpretation of the MS data depends on the reproducibility of the method employed. On line 152 the text states that permethylated glycan samples were dissolved in 10 microliter of solvent. What measures were taken to keep the glycan concentration constant among the samples? How many shots were acquired per summed MALDI MS spectrum? How many technical replicates were acquired? What is the reproducibility of the HCG purification, digestion, glycan release, workup method? What sample was used as a performance standard to verify consistent glycan quantification?

6. As stated in the abstract and elsewhere, the authors seem to use a p value threshold to identify glycan abundance differences that are significant. Either the p values or the significance threshold should be corrected for multiple comparisons. Please confirm the method used for correction.

Reviewer #2: The work describes the changes in glycosylation associated with a single protein, chorionic gonadotropin, during pregnancy in normal and a specific and rare condition. The protein is isolated using antibodies, and the N-glycans are removed and analyzed by MALDI -MS. The manuscript is generally acceptable, however there are a few issues that could be further clarified.

Here are the issues both minor and major that should be addressed.

1. MALDI is not generally sensitive at least compared to nanoESI so calling it “ultrasensitive MALDI” seems a little deceiciving. It doesn’t seem like they modified the instrument, so it’s a standard MALDI.

2. The sample size is really small. There were four subjects and one was dropped. The statistical analysis seems like an overkill, given the sample size. Some comment here regarding the sample size would be welcome.

3. What is the general CV of the MALDI method? There were no reproducibility studies performed or cited.

4. The structures are all mainly putative, and there is too much emphasis on the structures given that they are guesses. For example, can they confirm that these are really Lewis x structures? The structural guesses should be toned down a bit.

5. Similarly, the structure of the bisecting GlcNAc. There are some comments regarding transferases, but it’s not clear nor is it shown that these can determine the structure.

6. PLOS authors have the option to publish the peer review history of their article (what does this mean?). If published, this will include your full peer review and any attached files.

Reviewer #1: No

Reviewer #2: No

---

## [Author Response · Author response to Decision Letter 0]

17 Dec 2019

Insights into the hyperglycosylation of human chorionic gonadotropin revealed by glycomics analysis (PONE-D-19-25594)-Response to reviewers

Reviewer 1 : Specific points have been addressed as follows:

1) It would be very useful to show the measured proteomes of representative hCG samples. This would demonstrate the extent to which observed released glycans can be attributed to hCG, rather than co-isolated glycoproteins.

Representative proteomic data has been added to the supporting information section and cited in the manuscript. 

2) Please discuss the rationale for using MALDI TOF MS profiling of glycans. LC-MS has better dynamic range and would be the logical choice for quantitative studies.

The use of MALDI TOF MS for profiling of glycans is a highly effective methodology. Whilst we agree with the reviewer that in fields such as proteomics LC-MS has indeed been almost universely adopted as the MS method of choice in the glycan analytic field there is a greater diversity of MS methodology utilized. For example, a simple pubmed search with the terms MALDI-MS glycan 2019 gives 109 publications whilst LC-MS glycan 2019 gives 104 publications. We have also added the following text into the paper to allow a greater explanation of our MALDI-MS based methodology. 

“Τhe N-glycan profiles of all hCG samples derived from the above pregnancy conditions were determined using a glycomics strategy previously optimised for the characterisation of N-glycomes from a wide variety of samples including pregnancy associated glycoproteins, such as human glycodelin (41). This glycomic strategy is based on MALDI-MS and MS/MS analyses of the total population of N-glycans after their release from the polypeptide backbone of hCG via digestion with peptide N-glycosidase F. The glycans were permethylated prior to MALDI analysis in order to enhance sensitivity and to allow unambiguous MS/MS fragmentation. The MALDI-MS experiments defined glycan compositions whilst MS/MS experiments determined antennae sequences, as well as the location of fucose residues in Lewis structures”

3) How is “ultrasensitive” MALDI distinguished from traditional MALDI?

We have removed the term “ultrasensitive” throughout the manuscript.

4) It does not appear that the authors released O-glycans. Doing so would have allowed them to shed light on the extent to which hyperglycosylation of hCG is due to changes in O-glycosylation as mentioned on p. 3, line 74.

We agree with the reviewer that it would indeed have been advantageous to characterise both the N- and O-glycosylation of the hCG samples. However, sample quantities were very limited reflective of the challenges associated with obtaining the original starting biological material from pregnant women. This does not detract from our original objective which was to characterise the poorly understood N-glycosylation of pregnancy hCG. In so doing we have made important new discoveries. 

5) The quantitative interpretation of the MS data depends on the reproducibility of the method employed. On line 152 the text states that permethylated glycan samples were dissolved in 10 microliter of solvent. What measures were taken to keep the glycan concentration constant among the samples? How many shots were acquired per summed MALDI MS spectrum? How many technical replicates were acquired? What is the reproducibility of the HCG purification, digestion, glycan release, workup method? What sample was used as a performance standard to verify consistent glycan quantification?

Many of these questions are not as relevant in glycomic analysis compared to proteomic analysis. All relevant experimental details are available in the materials and methods section of the paper and the reference publications. Our glycan quantification relies on comparing relative abundance of ions within a spectrum. In glycomics experiments it has been demonstrated that quantitations based on signal intensities of permethylated glycans with in the MALDI-TOF MS spectrum is a reliable method (see Wada, Y., Azadi, P., Costello, C. E., Dell, A., Dwek, R. A., Geyer, H., Geyer, R., Kakehi, K., Karlsson, N. G., Kato, K., Kawasaki, N., Khoo, K. H., Kim, S., Kondo, A., Lattova, E., Mechref, Y., Miyoshi, E., Nakamura, K., Narimatsu, H., Novotny, M. V., Packer, N. H., Perreault, H., Peter-Katalinic, J., Pohlentz, G., Reinhold, V. N., Rudd, P. M., Suzuki, A., and Taniguchi, N. (2007) Glycobiology 17, 411-422).

6) As stated in the abstract and elsewhere, the authors seem to use a p value threshold to identify glycan abundance differences that are significant. Either the p values or the significance threshold should be corrected for multiple comparisons. Please confirm the method used for correction.

We have added the following text into the paper to allow a greater explanation.

“All statistical analyses were performed using SPSS version 25. A paired t-test was applied to analyse differences for paired hCG samples (EP-hCG versus LP-hCG), while an independent sample t-test was used for differences between EP-hCG and GTD-hCG. Significance was taken as P < 0.05. The significance threshold was corrected for multiple comparisons using Bonferroni method. For the paired t-test epitopes variables (LacNAc, NeuAc and LewisX), the corrected P was set at <0.0166, while for the N-glycans variable (high mannose, mono-, bi-, tri- and tetra-antennary N-glycans, agalactosylated, bisected and core-fucosylated N-glycans) it was set at <0.0063. For the independent sample t-test epitopes variables (LacNAc, NeuAc and LewisX), the corrected P was set at <0.0166, while for the N-glycans variable (mono-, tri- and tetra-antennary N-glycans, agalactosylated, bisected and core-fucosylated N-glycans) it was set at <0.0083. The Bejamini and Hochberg methods for false discovery rate was also applied. here The latter method was available in the SPSS software package through the syntax command following instructions published on the IBM support web page (Bejamini and Hochberg, document number 418001). The strictest result of the above methods was taken into account in terms of corrected significance. For the paired t-test, the test of normality with Shapiro-Wilk’s statistic and outliers was taken into consideration for the values resulting from the differences of the above variables (EP-hCG versus LP-hCG). For the independent sample t-test, the test of normality with Shapiro-Wilk’s statistic, outliers, and variance according to Levene’s test for equality of variances were taken into consideration. Variables not meeting the above assumptions were excluded from the analysis. 

Hierarchical cluster analysis was applied using between-groups linkage as a clustering method and Euclidean distance as a measure interval (similar results were obtained with other clustering methods and measure intervals). On EP-hCG versus GTD-hCG samples cluster analysis was performed for the variables found to be significant after correction for multiple comparisons using the criteria as stated above at least at the P < 0.05 level (tri-antennary, tetra-antennary, core fucosylation, bisected, LacNAc, LewisX, NeuAc)., while fFor the complete set of samples (EP-hCG, LP-hCG and GTD-hCG) hierarchical cluster analysis was performed using as variables the NeuAc and bisected N-glycans.”

Reviewer 2 : Specific points have been addressed as follows:

1) MALDI is not generally sensitive at least compared to nanoESI so calling it “ultrasensitive MALDI” seems a little deceiciving. It doesn’t seem like they modified the instrument, so it’s a standard MALDI.

We have removed the term “ultrasensitive” throughout the manuscript.

2) The sample size is really small. There were four subjects and one was dropped. The statistical analysis seems like an overkill, given the sample size. Some comment here regarding the sample size would be welcome.

Please see responses to points 4 and 6 of reviewer 1.

3) What is the general CV of the MALDI method? There were no reproducibility studies performed or cited.

Please see response to points 5 reviewer 1.

4) The structures are all mainly putative, and there is too much emphasis on the structures given that they are guesses. For example, can they confirm that these are really Lewis x structures? The structural guesses should be toned down a bit.

We apologies for not making it clearer how are glycan structures are defined. The structural assignments that we have presented are rigorous. To allow a greater understanding to the general reader we have added the following text into the paper to allow a greater explanation.

“Τhe N-glycan profiles of all hCG samples derived from the above pregnancy conditions were determined using a glycomics strategy previously optimised for the characterisation of N-glycomes from a wide variety of samples including pregnancy associated glycoproteins, such as human glycodelin (41). This glycomic strategy is based on MALDI-MS and MS/MS analyses of the total population of N-glycans after their release from the polypeptide backbone of hCG via digestion with peptide N-glycosidase F. The glycans were permethylated prior to MALDI analysis in order to enhance sensitivity and to allow unambiguous MS/MS fragmentation. The MALDI-MS experiments defined glycan compositions whilst MS/MS experiments determined antennae sequences, as well as the location of fucose residues in Lewis structures.”

5) Similarly, the structure of the bisecting GlcNAc. There are some comments regarding transferases, but it’s not clear nor is it shown that these can determine the structure.

We have added the following text into the paper to allow a greater explanation.

“Evidence for bisected glycans was obtained from observing whether glycans which had been shown by MS/MS to contain terminal GlcNAc residues were capped with a galactose residue after treatment with β-galactosyltransferase and UDP-Gal. Bisected GlcNAc residues are not accessible to β-galactosyltransferase and therefore are not capped with a galactose residue during this procedure. In contrast, terminal GlcNAc residues on truncated N-glycan antennae are fully accessible to the β-galactosyltransferase. Therefore, their capping with galactose serves as a positive control for β-galactosyltransferase activity. This methodology has been previously shown to be a reliable way of confirming bisected GlcNAc (38).”

---

## [Decision Letter · Decision Letter 1]

17 Jan 2020

Insights into the hyperglycosylation of human chorionic gonadotropin revealed by glycomics analysis

PONE-D-19-25594R1

Dear Dr. ibeto,

We are pleased to inform you that your manuscript has been judged scientifically suitable for publication and will be formally accepted for publication once it complies with all outstanding technical requirements.

With kind regards,

Roger Chammas, M.D, Ph.D

Academic Editor

PLOS ONE

Additional Editor Comments (optional):

Reviewers' comments:

Reviewer's Responses to Questions

**Comments to the Author**

1. If the authors have adequately addressed your comments raised in a previous round of review and you feel that this manuscript is now acceptable for publication, you may indicate that here to bypass the “Comments to the Author” section, enter your conflict of interest statement in the “Confidential to Editor” section, and submit your "Accept" recommendation.

Reviewer #1: (No Response)

2. Is the manuscript technically sound, and do the data support the conclusions?

Reviewer #1: Yes

3. Has the statistical analysis been performed appropriately and rigorously? 

Reviewer #1: Yes

4. Have the authors made all data underlying the findings in their manuscript fully available?

Reviewer #1: Yes

5. Is the manuscript presented in an intelligible fashion and written in standard English?

Reviewer #1: Yes

6. Review Comments to the Author

Reviewer #1: The authors have made detailed responses to the concerns raised in the first review. The responses show that the authors have taken the reviewer concerns seriously. Their responses were very thorough and therefore appropriate.

7. PLOS authors have the option to publish the peer review history of their article (what does this mean?). If published, this will include your full peer review and any attached files.

Reviewer #1: No

---

## [Editor Report · Acceptance letter]

29 Jan 2020

PONE-D-19-25594R1 

Insights into the hyperglycosylation of human chorionic gonadotropin revealed by glycomics analysis 

Dear Dr. ibeto:

I am pleased to inform you that your manuscript has been deemed suitable for publication in PLOS ONE. Congratulations! Your manuscript is now with our production department. 

With kind regards,

on behalf of

Prof. Roger Chammas 

Academic Editor

PLOS ONE